# Extraction of Alkaloids Using Ultrasound from Pulp and By-Products of Soursop Fruit (*Annona muricata* L.)

**Gabriela Aguilar-Hernández** [1], **Luis Gerardo Zepeda-Vallejo** [2,*],
**María de Lourdes García-Magaña** [1], **María de los Ángeles Vivar-Vera** [3],
**Alejandro Pérez-Larios** [4], **Manuel I. Girón-Pérez** [5], **Ana Velia Coria-Tellez** [6],
**Cristian Rodríguez-Aguayo** [7] **and Efigenia Montalvo-González** [1,*]

[1]  Laboratorio Integral de Investigación de Alimentos, Tecnológico Nacional de Mexico-Instituto Tecnológico de Tepic. Av. Tecnológico 2595, Lagos del Country, Tepic 63175, Nayarit, Mexico; gaby.mca2017@gmail.com (G.A.-H.); mgarciam@tepic.tecnm.mx (M.d.L.G.-M.)

[2]  Depto. de Química Orgánica, Escuela Nacional de Ciencias Biológicas, Instituto Politécnico Nacional, Av. Prolongación de Carpio y Plan de Ayala s/n, Col. Santo Tomas, Delegación Miguel Hidalgo, Ciudad de Mexico 11340, Mexico

[3]  Tecnológico Nacional de Mexico/Campus-Instituto Tecnológico de Tuxtepec, Depto. de Ingeniería Química y Bioquímica-Maestría en Ciencias en Alimentos, Av. Dr. Víctor Bravo Ahuja S/N, Col. 5 de Mayo, Tuxtepec 68350, Oaxaca, Mexico; mangeles@ittux.edu.mx

[4]  División de Ciencias Agropecuarias e Ingenierías, Centro Universitario de los Altos, Universidad de Guadalajara, Av. Rafael Casillas Aceves 1200, Tepatitlán 47600, Jalisco, Mexico; alarios@cualtos.udg.mx

[5]  Laboratorio de Inmunotoxicología, Secretaría de Investigación y Posgrado, Universidad Autónoma de Nayarit, Boulevard Tepic-Xalisco s/n, Cd. de la Cultura Amado Nervo, Tepic 63000, Nayarit, Mexico; ivangiron@uan.edu.mx

[6]  Laboratorio de Análisis y Diagnóstico del Patrimonio, El Colegio de Michoacán A.C., Cerro de Nahuatzen 85, Fracc. Jardines del Cerro Grande, La Piedad 59370, Michoacán, Mexico; anac@colmich.edu.mx

[7]  Department of Experimental Therapeutics, The University of Texas MD Anderson Cancer Center, Houston, TX 77030, USA; CRodriguez2@mdanderson.org

*  Correspondence: lzepeda@ipn.mx (L.G.Z.-V.); emontalvo@ittepic.edu.mx (E.M.-G.); Tel.: +52-311-211-94 (ext. 239) (E.M.-G.)

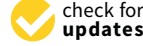

**Featured Application: Ultrasound assisted extraction (UEA) is a methodology considered ecological and low cost. The optimization of the UAE conditions to extract bioactive compounds, such as alkaloids, is useful for future applications in the industry. Alkaloids are compounds of interest to the pharmaceutical industry; a rich plant source of alkaloids is soursop by-products. It is of great importance since they are an alternative for its exploitation and its use will reduce environmental pollution.**

**Abstract:** The main goal of this work was to measure the total alkaloid content (TALC) from pulp, peel, seed, and columella of soursop fruit (*Annona muricata* L.) by ultrasound-assisted extraction (UAE) and to obtain the best conditions of the UAE with the response surface methodology (RSM). We evaluated the effect of amplitude (40%, 70%, and 100%), time (5, 10, and 15 min) and pulse-cycles (0.4, 0.7, and 1 s) and compared the best UAE conditions of alkaloids with a conventional extraction (maceration). The structural characterization of the raw material with the highest TALC was developed using nuclear magnetic resonance (NMR) techniques [$^1$H, $^{13}$C, heteronuclear single quantum correlation (HSQC), heteronuclear multiple bond correlation (HMBC), and homonuclear correlation spectroscopy (COSY)]. According to the RSM, the best conditions in the UAE for extracting alkaloids varied depending on the type of plant tissue. It took 5 min with an amplitude of 70% and pulse-cycles of 1,

0.4, and 1 s, respectively, to extract the highest TALC in peel, seed, and columella while the optimal conditions for extracting the largest amount of alkaloids from the pulp were obtained at 5 min in UAE with pulse-cycles of 0.55 s and 100% amplitude. The TALC was highest in the peel (7.48 mg/g), which was followed by the seed (2.31 mg/g), the pulp (1.20 mg/g), and the columella (0.79 mg/g) and was positively correlated ($R^2$ = 0.98–0.88) with the predicted values. In addition, the extraction alkaloids from the peel, pulp, seed, and columella using the UAE was 56.31, 5.45, 3.06, and 2.96 times higher, respectively, than the extraction by maceration. The alkaloids identified in the peel have not been reported and were nornuciferin, assimilobin, anonaine, and isolaureline. This study showed that the soursop fruit peel can be a source of alkaloids and that UAE has an important potential for extracting these compounds.

**Keywords:** *Annona muricata*; alkaloids; pulp; by-products; ultrasound-assisted extraction; 1D and 2D NMR experiments

---

## 1. Introduction

Soursop (*Annona muricata* L. or *A. muricata*) is a good source of bioactive compounds such as polyphenols, acetogenins, and alkaloids [1–4]. Polyphenols and acetogenins are widely investigated in the *A. muricata* plant. However, alkaloids (ALKs) from *A. muricata* by-products are little studied.

The biological activity of ALKs has been studied since the 19th century, and they have been used in medicine as anesthetics and analgesics. However, some plants are known to produce toxic ALKs. Therefore, it is necessary to determine the content and type of ALKs present in the herbs or plants to find their potential use in pharmacology [5]. Isoquinoline ALKs (assimilobin, nornuciferin, annonain, reticulin, and others) have been extracted from *A. muricata* leaves, roots, seeds, and pulp and these compounds have been shown to have analgesic, antidepressant, cytotoxic, antitumor, antineoplastic, antiplatelet, and dopaminergic activities [2,6–9]. These ALKs are involved in major behavioral disorders such as schizophrenia, Huntington's disease, attention deficit hyperactivity disorder, and Parkinson's disease. Precisely, alkaloids with (R)-apomorphine groups, aporfin semi-synthesized from morphine, have been used in the clinic for treating Parkinson's disease [10–12]. Likewise, it has been demonstrated that these alkaloids have an affinity for 5-HT1A receptors in vitro and participate in the biosynthesis of dopamine. It has been proposed that Annone-derived alkaloids could induce antidepressant type effects and cytotoxic activity [6,7].

However, there are no reports on the content and type of ALKs present in the other parts of *A. muricata* that are considered as by-products such as the peel or columella of the soursop fruit.

Ultrasound-assisted extraction (UAE) is a technology that has been widely used in recent years to extract bioactive compounds from different plants. The mechanism of action of the UAE is to produce cavitation that causes structural changes in the plant material by facilitating the extraction of compounds [13,14]. Ultrasound offers advantages that can improve extraction, reduce compound degradation, and processing time compared to heat extraction methods [15,16]. Ultrasonic cavitation allows for high diffusion of extraction solvents, higher mass transfer, and greater plant cell decomposition more than maceration, heating, and the soxhlet method (liquid-solid extraction) [13,14,17]. However, the utility or potential of the UAE depends on factors such as: the type of metabolite, the raw material, or the food matrix. Therefore, research on optimization of metabolite extraction by UAE should be established for each plant product [18,19].

The UAE has been applied for the extraction of ALKs from different plants. Dary et al. [20] used the UAE to extract three bioactive ALKs from *Stephania cambodica* and, with the response surface methodology (RSM), found the optimal extraction conditions (liquid-solid ratio 26.6:1 mL/g, ethanol 52%, 25 kHz, time 9 min, and 150 W). The authors concluded that the UAE is a practical extraction procedure for obtaining alkaloid-rich extracts for pharmacological research. Meanwhile,

Wang et al. [21] applied UAE and RSM to extract ALKs from *Sophora alopecuroides* seeds, and found that UAE improved yield by 32% to 136% compared to the enzymatic extraction method. On the other hand, Yohannes et al. [22] found that, when they used UAE (extraction time 30 min, 90 W) on *Radix physochlainae*, the extraction efficiency of ALKs was 121.3% compared to extraction by heating.

RSM is a tool to optimize the extraction processes of biological compounds because it considers the operating variables and their simultaneous interactions during the extraction of bioactive compounds from plant tissues [23]. Recently, Aguilar-Hernández [6,7] found that UAE was effective in extracting polyphenols and acetogenins from soursop pulp and by-products, and concluded that, depending on the type of bioactive compound extracted and on each part of the fruit, UAE extraction conditions change because factors such as the physicochemical stability of the compounds and the extraction matrix are involved. Therefore, the aim of this study was to extract total ALKs using ultrasound-assisted extraction (UAE) from the pulp, peel, seed, and columella of *A. muricata* to optimize extraction conditions with RSM and to characterize by using NMR techniques. The type of ALKs present in the raw material have the highest content of these compounds.

## 2. Materials and Methods

### 2.1. Plant Material

Soursop fruits (*A. muricata*) were harvested at consumption maturity (15–19 °Brix) in Compostela, Nayarit, Mexico. They were manually peeled to obtain the pulp and the by-products (peel, seeds, and columella). The raw materials were freeze-dried at −50 °C under pressure of 0.12 mbar (Labconco 77522020, Kansas City, MO, USA), then ground (Nutribullet NB-101B, Los Angeles, CA, USA), and sieved (500 μm, No. 35, Fisher Scientific, Hampton, NH, USA).

Quinine standard, acetic acid, Dragendorff's reagent, and methanol were purchased from Sigma-Aldrich (St. Louis, MO, USA).

### 2.2. Ultrasound-Assisted Extraction (UAE)

A Box-Behnken design was used for UAE. The individual effects and interactions of sonication amplitude ($X_{SA}$, %) at 40%, 70%, and 100%, extraction time ($X_{ET}$, 5, 10, and 15 min), and pulse cycles ($X_{PC}$, 0.4, 0.7, and 1 s) on total alkaloid content were evaluated in the experiment. To reduce the effect of unexplained variability on response variables, a randomized experimental order was used.

ALKs extraction from the by-products and pulp was performed by a UP400S ultrasonic system at 24 kHz frequency and 400 Watts (Hielscher Ultrasonics, Teltow, Germany). The ultrasonic probe (H7 Tip 7, Hielscher, Teltow, Germany) has a maximum amplitude of 100% or 175 μm and acoustic power density of 300 W/cm$^2$. This device was immersed in the extractant solution (2 cm). The freeze-dried samples (2 g) were placed in an extraction tube with methanol (15 mL), and treated under UAE controlled conditions (according to Box-Behnken design) at 25 ± 2 °C. A water recirculation bath (Thermo Scientific 2870, Waltham, MA, USA) was used to maintain the temperature. The samples were then centrifuged at 8000× g, 4 °C, and 10 min (Hermle Z32HK, Wehingen, Germany). The supernatants were recovered for analysis.

### 2.3. Total Alkaloid Content (TALC)

The TALC was determined according to the procedure described by Ontiveros-Rodriguez et al. [24] with modifications. The methanolic extracts obtained were diluted with 10 mL of methanol, and 5 mL of 5% HCl were added. The samples were shaken moderately for 4 h at room temperature (25 ± 5 °C) and then left to rest in refrigeration (4 °C) overnight. The extracts were filtered, placed on ice, and their pH was adjusted to 10.0 with 15% NaOH. Lastly, $CH_2Cl_2$ in 1:2 ratio (extract: $CH_2Cl_2$) was added to the samples, and the organic layer was recovered. This step was performed two more times to bring together the organic layers obtained. The extracts (100 μL) were mixed with the Dragendorff's reagent (1000 μL) and the absorbance was measured at 530 nm in a spectrophotometer (Jenway 6705, Dunmow,

UK). The TALC was calculated with a standard quinine curve, and the results were expressed as equivalent milligrams of quinine per gram of dry weight (mg/g DW).

## 2.4. Response Surface Methodology Analysis

The RSM was applied to the TALC data to obtain the best extraction conditions for each raw material. To calculate the predicted response, a second-order polynomial equation was used (Equation (1)).

$$Y = \beta 0 + \sum_{i=A}^{E} \beta i \, X i + \sum_{i=A}^{E} \sum_{j=A \neq i}^{E} \beta ij \, X i + E \tag{1}$$

where Y is the predicted response (TALC), $\beta_0$ is a constant, $\beta_i$ are the main effect coefficient for each variable, Xi is the coded or uncoded values for the factors ($X_{ET}$, $X_{PC}$, and $X_{SA}$), and βij are the coefficients of the interaction effect. To determine the main effects, interactions and suitability of the model, an analysis of variance (ANOVA) was performed and the coefficients of determination (R-squared and R-adjusted) were quantified; the test of non-adjustment was also used with a significance level represented by 1%.

## 2.5. Model Reliability of Ultrasound-Assisted Extraction (UAE)

To verify model accuracy, the optimal UAE conditions obtained from RSM analysis were used to extract TALC from pulp and by-products. These latter results were compared with the TALC obtained by maceration. The effectiveness of UAE (%) was calculated.

### 2.5.1. Extraction by Maceration (ME) of Alkaloids

Extraction by maceration was carried out with 2 g of freeze-dried sample in 15 mL of methanol at room temperature for one week [24]. After maceration, the methanolic extracts were treated and quantified, as described in Section 2.3.

### 2.5.2. Effectiveness of UAE in the Alkaloid Extraction

The effectiveness of alkaloid extraction was calculated as the ratio between TALC obtained by UAE and TALC obtained by maceration (Equation (2)).

$$\text{Effectiveness (n} - \text{times)} = \frac{\text{TALC by UAE}}{\text{TALC by maceration}} \tag{2}$$

## 2.6. Structural Identification of Alkaloids

The raw material with the highest ALKs content derived from results was selected in the first stage. The dried powder (700 g) of the selected raw material was added to methanol, and the ALKs were extracted under the optimal UAE conditions. The crude extract was treated as in Section 2.3 to obtain an organic crude extract, and concentrated in a rotary evaporator to produce 2.5 g of dried alkaloidal extract. The alkaloidal extract was eluted on a chromatographic column (4.46 × 60 cm, 80 g SiO$_2$ 60 mesh) with CH$_2$Cl$_2$/AcOEt/EtOH (1:2:0.2 *v/v*). A total of 115 fractions were obtained. Each fraction was performed to thin-layer chromatography (Analytical Chromatography, HX312859, Darmstadt, Germany). The ALKs were visualized with Dragendorff´s reagent [24].

The alkaloidal positive fractions were analyzed by $^1$H and $^{13}$C nuclear magnetic resonance (NMR) spectroscopy in 1D and 2D. Pre-purified ALKs were extensively analyzed by $^1$H and $^{13}$C NMR spectroscopy in 1D and 2D. The samples (10 mg) were dissolved in 0.5 mL of deuterated chloroform (CDCl$_3$), and was then poured into a magnetic resonance tube (5 mm diameter). The $^1$H and $^{13}$C NMR and homo and heteronuclear in 1D and 2D spectra were measured in an NMR spectrometer (Varian VNMR System, Walnut Creek, CA, USA) at 500 MHz. The chemical shifts were measured in ppm or δ.

*2.7. Statistical Analysis*

Data were expressed as the mean ± standard deviation ($n = 3$). The RSM was used as well as an analysis of variance (ANOVA) ($p < 0.05$) with the STATISTICA v.10 software (Statsoft, Tulsa, OK, USA). To analyze the significant difference between means, the Fisher test was used ($\alpha = 0.05$).

## 3. Results and Discussions

*3.1. Ultrasound-Assisted Extraction (UAE) of Total Alkaloids Content (TALC) in Soursop Samples*

The experimental design and results obtained at the TALC are shown in Table 1. The UAE treatments and raw material had significant effects as the TALC of the pulp and by-products (peel, seeds, and columella) depended on the experimental conditions in the UAE ($p < 0.05$). The highest TALC (7.48, 2.31, and 0.60 mg/g, respectively) was found in the peel, seed, and columella under the same conditions of time (5 min) and sonication amplitude (70%) with 1-s pulse-cycle for peel and columella, and 0.4-s pulse-cycle for seeds. Likewise, the highest TALC in the pulp (1.16 mg/g) was found using 5 min of extraction, 0.7 s of pulse-cycles, and 100% amplitude. These differences in TALC could be attributed to different factors. First, the effects of UAE conditions and structural composition of raw materials. The extraction of bioactive compounds by ultrasound is achieved by the vibration of the ultrasonic waves, which cause the acceleration of the particles and allows the solute to pass quickly to the solvent. This increases its diffusivity [14]. Some authors argue that UAE produces mechanical effects that modify the structure of the cell wall and it becomes more porous when facilitating extraction [25].

**Table 1.** Effect of the variables of the process of extraction assisted with ultrasound on the content of total alkaloids from A. muricata peel, seed, columella, and pulp.

| Treatment | Ultrasound Conditions | | | Total Alkaloids (mg/g Dry Weight) | | | |
|---|---|---|---|---|---|---|---|
| | $^1X_{ET}$ | $^2X_{SA}$ | $^3X_{PC}$ | Peel | Seed | Columella | Pulp |
| 1 | 5 | 40 | 0.7 | 3.03 ± 0.02 [jA] | 0.93 ± 0.01 [kB] | 0.51 ± 0.01 [bC] | 0.12 ± 0.01 [kD] |
| 2 | 15 | 40 | 0.7 | 2.64 ± 0.01 [lA] | 0.91 ± 0.01 [kB] | 0.48 ± 0.02 [dC] | 0.46 ± 0.01 [hD] |
| 3 | 5 | 100 | 0.7 | 3.26 ± 0.03 [hA] | 1.86 ± 0.01 [cB] | 0.15 ± 0.02 [iD] | 1.16 ± 0.02 [aC] |
| 4 | 15 | 100 | 0.7 | 4.74 ± 0.01 [cA] | 0.81 ± 0.02 [mB] | 0.41 ± 0.01 [fC] | 0.20 ± 0.01 [jD] |
| 5 | 5 | 70 | 0.4 | 3.11 ± 0.02 [iA] | 2.31 ± 0.02 [aB] | 0.40 ± 0.02 [fD] | 0.95 ± 0.01 [bC] |
| 6 | 15 | 70 | 0.4 | 2.80 ± 0.01 [kA] | 1.96 ± 0.01 [bB] | 0.22 ± 0.01 [gD] | 0.76 ± 0.01 [eC] |
| 7 | 5 | 70 | 1 | 7.48 ± 0.01 [aA] | 1.18 ± 0.01 [fB] | 0.60 ± 0.01 [aC] | 0.43 ± 0.01 [iD] |
| 8 | 15 | 70 | 1 | 7.11 ± 0.01 [bA] | 1.18 ± 0.01 [fgB] | 0.50 ± 0.01 [bD] | 0.65 ± 0.01 [fC] |
| 9 | 10 | 40 | 0.4 | 3.57 ± 0.01 [fA] | 1.17 ± 0.01 [gB] | 0.22 ± 0.01 [ghD] | 0.75 ± 0.01 [eC] |
| 10 | 10 | 100 | 0.4 | 4.06 ± 0.01 [dA] | 1.01 ± 0.01 [iB] | 0.04 ± 0.02 [jD] | 0.76 ± 0.01 [eC] |
| 11 | 10 | 40 | 1 | 3.88 ± 0.01 [eA] | 1.52 ± 0.02 [dB] | 0.05 ± 0.02 [jD] | 0.88 ± 0.01 [cC] |
| 12 | 10 | 100 | 1 | 3.56 ± 0.01 [fA] | 1.00 ± 0.02 [jB] | 0.41 ± 0.03 [fD] | 0.65 ± 0.01 [fC] |
| 13 | 10 | 70 | 0.7 | 2.60 ± 0.01 [mA] | 1.87 ± 0.01 [cB] | 0.46 ± 0.01 [eD] | 0.82 ± 0.01 [dC] |
| 14 | 10 | 70 | 0.7 | 3.11 ± 0.01 [iA] | 1.06 ± 0.02 [hB] | 0.15 ± 0.01 [iD] | 0.63 ± 0.01 [gC] |
| 15 | 10 | 70 | 0.7 | 3.51 ± 0.01 [gA] | 1.36 ± 0.01 [eB] | 0.21 ± 0.01 [hD] | 0.81 ± 0.02 [dC] |

$^1X_{ET}$: extraction time (min). $^2X_{SA}$: sonication amplitude (%). $^3X_{PC}$: pulses-cycles. Data are expressed as the mean ± SD ($n = 3$). Same capital letters indicate no significant statistical differences between the raw materials ($p < 0.05$) and same lowercase letters indicate no significant statistical differences between treatments ($p < 0.05$).

The differences in the conditions of UAE for extracting these bioactive compounds depended on the physical-mechanical phenomenon that each UAE combination generates in each raw material influencing their structural composition. The longer treatment time, the highest ultrasonic amplitude or density of acoustic energy, and a constant acoustic irradiation cause a high cavitation, which produces to greater cellular lysis [14]. However, it has also been shown that extreme ultrasound conditions can decrease the content of bioactive compounds by the phenomenon called sonolysis, which causes the formation of hydroxyl radicals that cause oxidation [26]. This may indicate that the TALC extracted from each *A. muricata* raw material depends on the greater or lesser susceptibility to cell disruption and the stability of the ALK at the higher UAE extraction conditions of 100% amplitude, 1-s pulse cycle (permanent acoustic irradiation), and 15 min. Second, the difference in the ALKs content depended

on the synthesis of them in each raw material. Three important functions of alkaloids in plants are known such as defensive route as toxic compounds for animals and humans, and hormonal route because they function as phytohormones in some plants and an allelopathic route, which influence the growth of other organisms [8,27]. Therefore, the highest TALC in peel and seed was likely due to these organs increasing the biosynthesis of ALKs as a defense system because the function of the peel is to protect the fruit and the seed to the embryo [28]. It is important to point out that some research studies demonstrated that the UAE could be the most efficient method for extracting ALKs in different plants [20,29,30].

Teng and Choi [31] reported that the extraction efficiency of ALKs from *Rhizoma coptidis* was dependent on time, temperature, and solvent concentration. Dary et al. [20] showed that, when ultrasound extraction is applied in *Stephania cambodica*, the short times and low extraction temperatures have a significant effect on TALC because the UAE prevents the degradation of thermolabile compounds and the possible co-extraction of components undesirable in plant material. Similarly, Hossain et al. [29] found that, when applying UAE to obtain ALKs in potato peel, the extraction efficiency is mainly influenced by the amplitude. In this investigation, optimal UAE conditions to extract ALKs varied according to the matrix of each raw material, likely due to differences in chemical composition of the cell matrix, and types of ALKs present in them.

### 3.2. Response Surface Methodology (RSM) Analysis

The UAE conditions on TALC from raw materials are shown in Figure 1A (peel), Figure 1B (seed), Figure 1C (columella), and Figure 1D (pulp). There is a significant interaction effect between the factors evaluated ($p < 0.05$). The elliptical shape of the plots indicates the significant effect of interactions on the TALC [17]. All the conditions of the UAE caused an ALK extraction in all raw materials. The effect (negative or positive) of the factors on the extraction of ALKs depended on each matrix, and the significant effect of the independent variables (and their interactions) on each raw material was as follows: for peel, $X_{PC} > X_{ET} > X_{AS}$, seed $X_{ET} > X_{AS} > X_{PC}$, columella, $X_{ET} > X_{PC} > X_{AS}$, and pulp, $X_{ET} > X_{PC} > X_{SA}$.

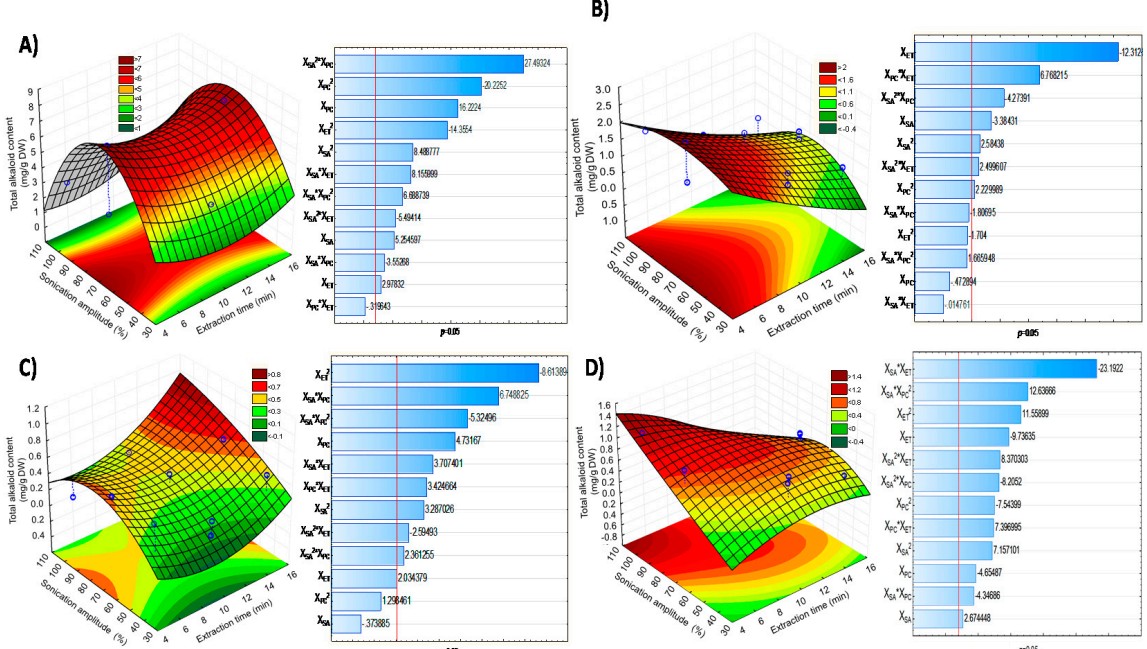

**Figure 1.** Response surface plots and pareto charts of total alkaloid content from *A. muricata* peel (**A**), seed (**B**), columella (**C**), and pulp (**D**) after the ultrasound-assisted extraction. DW = dry weight. $X_{SA}$ = sonication amplitude (%). $X_{ET}$ = extraction time (min). $X_{PC}$ = pulse cycle.

On the other hand, RSM was performed using multiple regression techniques. Second order polynomial equations (Table 2) were obtained to calculate the predicted TALC for pulp, peel columella, and seeds, which describe the effect of the test factors as well as the effect of all interactions on the response variable. Regression analysis showed that the TALC experimental data from peel ($R^2 = 0.98$), seed ($R^2 = 0.88$), columella ($R^2 = 0.88$), and pulp ($R^2 = 0.97$) have a reasonable correlation and adequate adjustment with the model (Lack of fit, $p > 0.05$) (Table 2). The lack of fit ensures the fitness of the model, and this demonstrates the approximation to a real system.

**Table 2.** Predicted mathematical model for the extraction of total alkaloids (mg/g DW) from the peel, seed, columella, and pulp from *Annona muricata* after the UAE.

| Raw Material | * Polynomial Equation | $R^2$ |
|---|---|---|
| Peel | $35.3989 - 0.6829X_{SA} + 0.0042\ X_{SA}{}^2 - 69.6346X_{PC} + 27.3493X_{PC}{}^2 - 0.4419X_{ET} + 0.0341X_{ET}{}^2 + 1.4082\ X_{SA}{}^*X_{PC} - 0.20\ X_{SA}{}^*X_{PC}{}^2 - 0.0082\ X_{SA}{}^{2*}X_{PC} - 0.0107\ X_{SA}{}^*X_{ET} + 0.001\ X_{SA}{}^{2*}X_{ET} - 0.0122X_{PC}{}^*X_{ET}$ | 0.98 |
| Seeds | $2.4056 + 0.0607\ X_{SA} - 0.0005X_{PC}{}^2 + 0.7420X_{PC} + 1.7927X_{PC}{}^2 - 0.497X_{ET} + 0.0036X_{ET}{}^2 - 0.1073X_{SA}{}^*X_{PC} - 0.0443\ X_{SA}{}^*X_{PC}{}^2 + 0.00110\ X_{SA}{}^{2*}X_{PC} + 0.0055\ X_{SA}{}^*X_{ET} - 0.00\ X_{SA}{}^{2*}X_{ET} - 0.23X_{PC}{}^*X_{ET}$ | 0.88 |
| Columella | $-0.66319 + 0.03356\ X_{SA} - 0.00018\ X_{SA}{}^2 + 3.93642X_{PC} - 4.2610X_{PC}{}^2 - 0.06707X_{ET} + 0.00727X_{TE}{}^2 - 0.02885\ X_{SA}{}^*X_{PC} + 0.05652\ X_{SA}{}^*X_{PC}{}^2 - 0.00025\ X_{SA}{}^{2*}X_{PC} - 0.00339\ X_{SA}{}^*X_{ET} + 0.00003\ X_{SA}{}^{2*}X_{ET} + 0.01257X_{PC}{}^*X_{ET}$ | 0.88 |
| Pulp | $1.5584 + 0.01X_{SA} - 0.00\ X_{SA}{}^2 - 8.554X_{PC} + 7.682X_{PC}{}^2 + 0.06X_{ET} - 0.0067X_{ET}{}^2 + 0.0386\ X_{SA}{}^*X_{PC} - 0.0923\ X_{SA}{}^*X_{PC}{}^2 + 0.001\ X_{SA}{}^{2*}X_{PC} + 0.003\ X_{SA}{}^*X_{ET} - 0.00\ X_{SA}{}^{2*}X_{ET} + 0.0688X_{PC}{}^*X_{ET}$ | 0.97 |

* $X_{SA}$ = Sonication amplitude (%). $X_{PC}$ = Pulses-cycles (s). $X_{TE}$ = Extraction time (min). $R^2$ = Regression coefficient.

Similar results were shown by Dary et al. [20] in the optimization of UAE of ALKs from *Stephania cambodica*. The extraction of ALKs from peel, seed, columella, and pulp can be calculated with predicted mathematical models (Table 2). However, some coefficients (Table 3) were not significant ($p > 0.05$) for the model. ALKs from peel, seed, columella, and pulp can be calculated with predicted mathematical models (Table 2).

**Table 3.** Regression coefficients of the predicted quadratic polynomial models with the conditions of ultrasound-assisted extraction on the total alkaloid content from *A. muricata* peel, seed, columella, and pulp.

| | Regression Coefficients | | | |
|---|---|---|---|---|
| **Fuente** | **CAT β-Coefficient** | | | |
| | **Peel** | **Seeds** | **Columella** | **Pulp** |
| Media/interception | 35.3989 | 2.4056 | −0.66319 | 1.5584 |
| $X_{SA}$ | −0.6829 * | 0.0607 ** | 0.03356 * | 0.0111 ** |
| $X_{SA}{}^2$ | 0.0042 * | −0.0005 * | −0.00018 ** | −0.0002 * |
| $X_{PC}$ | −69.6346 * | 0.7420 ** | 3.93642 * | −8.5540 * |
| $X_{PC}{}^2$ | 27.3493 * | 1.7927 ** | −4.2610 * | 7.6820 * |
| $X_{ET}$ | −0.4419 * | −0.497 * | −0.06707 * | 0.0591 * |
| $X_{ET}{}^2$ | 0.0341 * | 0.0036 ** | 0.00727 * | −0.0067 * |
| $X_{SA}{}^*X_{PC}$ | 1.4082 * | −0.1073 ** | −0.02885 ** | 0.0386 * |
| $X_{SA}{}^*X_{PC}{}^2$ | −0.20 * | −0.0443 ** | 0.05652 ** | −0.0923 * |
| $X_{SA}{}^{2*}X_{PC}$ | −0.0082 * | 0.0011 * | −0.00025 * | 0.0006 * |
| $X_{SA}{}^*X_{ET}$ | −0.0107 * | 0.0055 * | −0.00339 * | 0.003 * |
| $X_{SA}{}^{2*}X_{ET}$ | 0.0001 * | −0.00 * | 0.00003 * | −0.00 * |
| $X_{PC}{}^*X_{ET}$ | −0.0122 ** | 0.2292 * | 0.01257 ** | 0.069 * |
| R-square | 0.98 | 0.88 | 0.88 | 0.97 |
| R-adjusted | 0.98 | 0.84 | 0.84 | 0.96 |

$X_{ET}$ = Extraction Time. $X_{PC}$ = Pulses-cycles. $X_{SA}$ = Sonication amplitude. * Significant ($p < 0.05$). ** Non-significant ($p > 0.05$).

### 3.3. Model Reliability Assessment of Ultrasound-Assisted Extraction (UAE)

The optimal UAE conditions for the extraction of alkaloids from the peel, seed, columella, and pulp are shown in Table 4. Treatments were carried out using the best conditions for each raw material, as suggested by Aydar et al. [19] to modify the reliability of the model. The experimental data of TALC (Table 1) for all raw materials are consistent with the predicted data (Table 4).

**Table 4.** Optimal conditions of ultrasound-assisted extraction in total alkaloids from *A. muricata* peel, seed, columella, and pulp.

| Parameters | Peel | Seeds | Columella | Pulp |
|---|---|---|---|---|
| Extraction time (min) | 5 | 5 | 5 | 5 |
| Pulses-cycles (s) | 1 | 0.4 | 1 | 0.55 |
| Sonication amplitude (%) | 70 | 70 | 70 | 100 |
| Optimal response (mg/g DW) | 7.48 | 2.31 | 0.74 | 1.20 |
| −95% Confidence limit | 7.24 | 2.11 | 0.63 | 1.14 |
| +95% Confidence limit | 7.71 | 2.52 | 0.85 | 1.26 |

Confidence interval, the difference between upper and lower limits. −95% confidence limit, lower limit. +95% confidence limit, upper limit. DW = dry weight.

Thus, the use of UAE is useful to obtain an efficient extraction of alkaloids from *A. muricata*. RSM tools have been used to optimize the extraction of ALKs using UAE in tubers of the *Stephania cambodica* plant [20] and *Sophora alopecuroides* [21].

In all *A. muricata* raw materials, 5 min of UAE was sufficient to enhance TALC. However, the intensity of the ultrasound (70% amplitude) was the same for peel and columella with a 1-s pulse-cycle and seed with a 0.4-s pulse-cycle while 100% amplitude was necessary for the pulp with a 0.5-s pulse-cycle. According to Hielscher's ultrasound technology and the characteristics of ultrasound equipment, a 1-s pulse-cycle means a continuous acoustic irradiation with a 0.7-s pulse-cycle is 0.7 s of acoustic irradiation and 0.3 s is kept in pause and the 0.4-s pulse-cycle are 0.4 s of acoustic irradiation and 0.6 s in pause. Therefore, low cavitation was necessary to extract ALKs from the pulp or seeds likely because of the low content found.

TALC was significantly higher ($p < 0.05$) when ultrasound extraction was used than the alkaloid content obtained by maceration extraction (Table 5). The effectiveness of UAE was 56.3, 30.6, and 29.6 times higher for the peel, pulp, seed, and columella, respectively, when compared to maceration. Vinatoru et al. [32] reported an increase in the percentage (77%) of alkaloid extraction from ipecacuana root (*Cephaelis ipecacuanha*) and Jaborandi leaf (*Pilocarpus microphyllus*) when the extracts were obtained using UAE compared to the extraction using Soxhlet (22%). Additionally, Wang et al. [21] reported that alkaloid extraction from *Sophora alopecuroides* seeds produced the highest extraction yield (4.6% and 38.3%) when compared to extraction using reflux heating and enzymatic extraction, respectively.

### 3.4. Structural Identification of Alkaloids

Analysis of Nuclear Magnetic Resonance (NMR) Spectra Obtained from the Selective Extract of Alkaloids from the Peel of the Soursop Fruit

The peel from soursop was selected to obtain alkaloidal extract due to the highest content of them. The separation of the above extract by column chromatography gave a total of 180 fractions from which fractions 105 to 150 showed an alkaloid positive test when visualized with Dragendorff's reagent. It was possible to perform the structural identification of these compounds without further purification, using mainly proton and carbon-13 nuclear magnetic resonance ([1]H and [13]C NMR).

**Table 5.** Total alkaloid content and yield in peel, seed, columella, and pulp of *Annona muricata* using the optimal conditions of ultrasonic-assisted extraction and conventional extraction.

| Parameter/Method | UAE | CE | UAE | CE | UAE | CE | UAE | CE |
|---|---|---|---|---|---|---|---|---|
| | [1] Peel | | [2] Seeds | | [3] Columella | | [4] Pulp | |
| Total alkaloid content (mg/g DW) | $7.32 \pm 0.02$ [a] | $0.13 \pm 0.01$ [c] | $2.02 \pm 0.01$ [d] | $0.66 \pm 0.00$ [d] | $0.83 \pm 0.01$ [b] | $0.28 \pm 0.01$ [c] | $1.09 \pm 0.01$ [d] | $0.20 \pm 0.00$ [d] |
| Effectiveness UAE (n-times) | 56.31 | | 3.06 | | 2.96 | | 5.45 | |

All values are the means ± standard deviation of three determinations. Different letters in each row indicate significant statistical difference between treatments ($p < 0.05$). CE = Conventional extraction. Ultrasound-assisted extraction (UAE) experimental conditions. (1) $X_{ET} = 5$ min, $X_{PC} = 1$ s, $X_{SA} = 70\%$. (2) $X_{ET} = 5$ min, $X_{PC} = 0.4$ s, $X_{SA} = 70\%$; (3) $X_{ET} = 15$ min, $X_{PC} = 1$ s, $X_{SA} = 100\%$. (4) $X_{ET} = 5$ min, $X_{PC} = 0.55$ s, $X_{SA} = 100\%$.

Figure 2 shows the $^1$H NMR spectrum (bottom) of this set of fractions (F105–F150), which highlights the residual signal of the $CH_2O$ group of ethanol used in the washing process of glass material during the analysis as well as the $CHCl_3$ signal normally present as a reference signal in deuterated chloroform ($CDCl_3$) used as solvent for NMR analysis. The rest of the signals belong to aromatic protons, methoxy groups (OMe), and protons attached to $sp^3$ carbons (signals appearing between 2.50 and 4.30 ppm) (Figure 2). In the same figure is shown an expansion between 5.90 and 8.45 ppm, where the chemical shifts, integral sections (area under the curve), and multiplicity for the signals of aromatic and dioxolane protons can be appreciated in greater detail. The assignment of the $^1$H NMR spectrum was supported by establishing correlations of vicinal ($^3J$) and geminal ($^2J$) protons in the homonuclear correlation spectrum (COSY) (Figure 2, top) from which adjacent scalar-coupled protons could be identified. For instance, correlations between protons H-8 and H-9 of aporphines 3 and 4 was established through their respective cross-peaks correlations, as shown in Figure 2 (top). Following a similar analysis for the rest of the protons, it was possible to recognize the signals of four major compounds with the core structure of aporphine alkaloids: nornuciferine 1, asimilobine 2, anonaine 3, and isolaureline 4 (Figure 3).

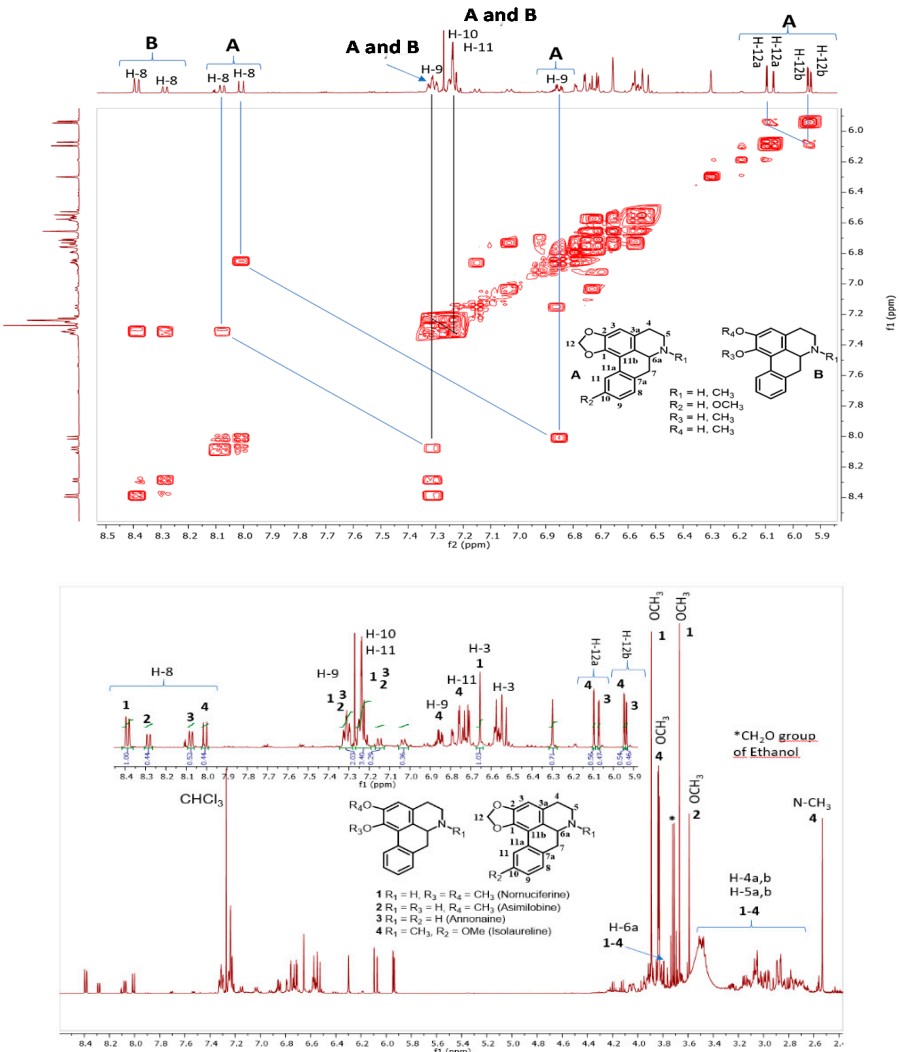

**Figure 2.** $^1$H nuclear magnetic resonance spectrum (**bottom**) and homonuclear correlation spectroscopy (COSY) (**top**) spectrum of the alkaloid-enriched F105-150 fractions obtained from the chromatography of the selective extract of alkaloids from *A. muricata* peel. Cross-correlation contour peaks shown in the COSY experiment (top) allowed to establish vicinal ($^3J$) and geminal ($^2J$) $^1$H-$^1$H connectivity, which concludes that the alkaloid-enriched fraction mainly comprises two pair of aporphine alkaloids (**A,B**).

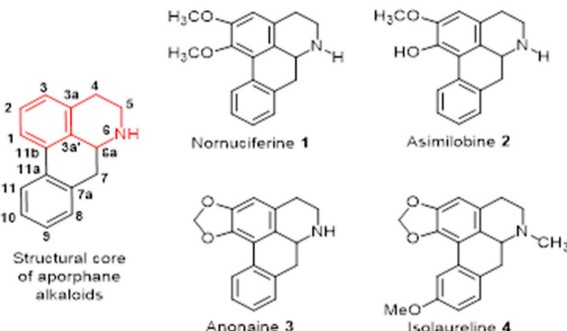

**Figure 3.** Basic structure of aporphine alkaloids with its systematic numbering (**left**), present in several species of genus *Annona*. The isoquinoline nucleus is shown in red. Structures of the major alkaloids (**right**) identified in alkaloid-enriched fractions F105-F150 obtained from the fruit of *Annona muricata*.

The most significant signals showing the presence of these four major alkaloids are described as follows. Between 7.95 to 8.45 ppm can be observed. The signals belong to proton H-8 of these derivatives. Similarly, the signals between 7.20 and 7.35 ppm of the aromatic protons H-9, H-10, and H-11 are grouped, while, in the region of 6.30 to 6.80 ppm, the singlet signals of proton H-3 are observed. The presence of protons for the $CH_2$ group of the dioxolane ring, assigned as H-12a and H-12b, are also very noticeable between 5.90 and 6.15 ppm. In turn, the singlet signals of methoxy groups ($OCH_3$) appear at a higher field between 3.50 and 3.90 ppm. In the same region, the H-6a signal usually appears, which, in this case, is hidden by the rest of the signals. The protons of the piperidine ring (H-4a,b, H-5a,b, and H-6a) appear as complex multiplicity signals in the region between 2.60 and 3.60 ppm. Lastly, in several aporphine alkaloids, there is sometimes a $CH_3$ group directly attached to nitrogen, which appears as a simple signal between 2.30 and 2.60 pm region in which one of these groups is observed (2.54 ppm).

The upper trace of the [1]H NMR spectrum (bottom) shows the signals for aromatic protons corresponding to nornuciferine 1, assimilobine 2, and anonaine 3 (H-3, H-8, H-9, H-10, and H-11) and isolaureline 4 (H-3, H-8, H-9, and H-11), as well as the protons for dioxolane ring (H-12a and H-12b) of anonaine 3 and isolaureline 4. Bold numbers identify the signals corresponding to each alkaloid according to its identification number.

It can be observed that the four double signals (due to scalar coupling with H-9, *J* = 8.7 Hz) belonging to H-8 provided clear evidence that four major alkaloids were present in this mixture. The integral values under each double signal represent the relative ratio between them, which gives 100:44:52:44 from left to right.

Complementarily, the four double signals (*J* = 1.4 Hz) shown between 5.90 and 6.15 ppm are typical of geminal protons (H-12a, H-12b) of the dioxolane rings (Figure 2, top, dioxolane derivatives A). A pair of these double signals belong to one alkaloid and the remaining pair belong to another one. Judging by their integral values, both at an averaged ratio of 55:45, it can be deduced that belonging to the same alkaloid gives the double signals appearing in 8.08 (integral 0.53) and 8.01 ppm (integral 0.44) of H-8, respectively. Therefore, it can be concluded that the signals appearing in 8.08, 6.10, and 5.95 ppm belong to protons H-8, H-12a, and H-12b, respectively, of an alkaloid with dioxolane ring (Figure 2, top, A structures) while the signals appearing in 8.01, 6.07, and 5.93 ppm belong to H-8, H-12a, and H-12b, respectively, of the second alkaloid with dioxolane ring. According to this analysis, it can be deduced that the double signals that appear in 8.39 and 8.28 ppm belong to H-8 of two aporphine derivatives without a dioxolane ring (Figure 2, top, B structures).

It can also be observed in Figure 4 where the [13]C NMR spectrum (bottom) shows the set of signals for aromatic carbons (δ 105–160), acetalic carbons of the dioxolane ring (δ 100.7–100.8), C-6a and $OCH_3$ groups (δ 53–65), and in a higher field (δ 28–47) N-$CH_3$, C-5, and C-4 carbons. The heteronuclear single quantum correlation (HSQC) experiment allowed to stablish de one-bond ([1]*J*) heteronuclear correlation between protons attached to their respective carbons (Figure 3, top). Both experiments complement

the spectroscopic evidence that the alkaloid-enriched fractions contain the presence of the following four compounds: nornuciferine 1, asimilobine 2, anonaine 3, and isolaureline 4. As mentioned above, such compounds possess the core structure of tetracyclic alkaloids named aporphine, which are biogenetically referred to as isoquinoline derivatives.

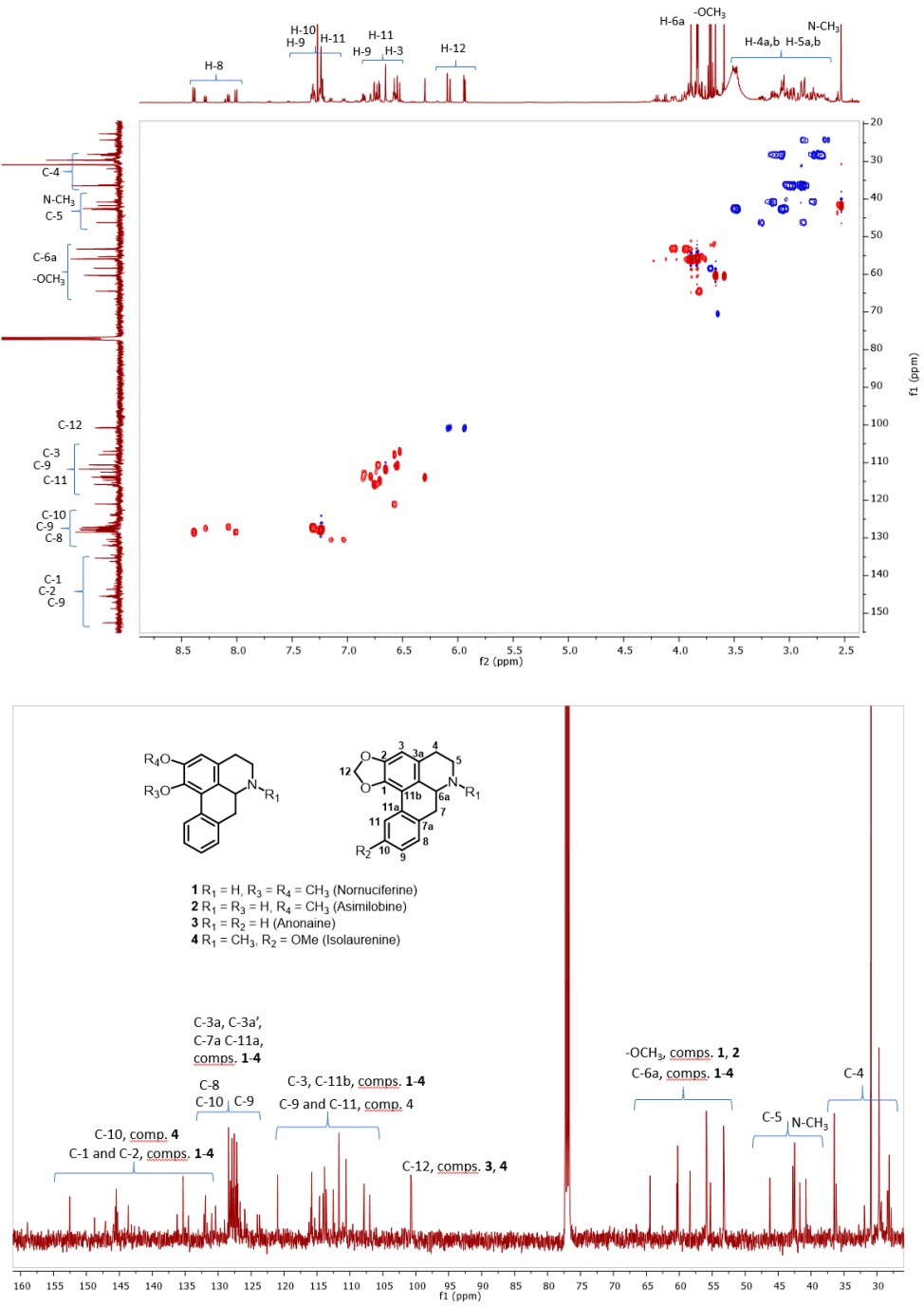

**Figure 4.** Heteronuclear single quantum correlation (HSQC, **top**) and [13]C nuclear magnetic resonance (NMR) spectrum (**bottom**) of the alkaloid-enriched F105-150 fractions obtained from the chromatography of the selective extract of alkaloids from *A. muricata* peel. HSQC experiment shows one-bond correlations for all protonated carbons and help discriminate all non-protonated carbons. [13]C NMR spectra (**bottom**) shows the range of chemical shifts in which the different types of carbons belonging to alkaloids nornuciferine 1, asimilobine 2, anonaine 3, and isolaureline 4 appear. Bold numbers identify the signals corresponding to each alkaloid according to its identification number.

Once the typical NMR signals of aporphine alkaloids were recognized, a bibliographic review was made on alkaloids present in the fruit of *A. muricata* in order to compare the information of our spectra with those described in the literature. With these considerations in mind, we proceeded to review the NMR data described for aporphine alkaloids isolated from the *A. muricata* fruit, or from other parts of this plant, including other species. Thus, Hasrat et al. [6,7] described the isolation of alkaloids nornuciferine 1, asimilobine 2, and anonaine 3 from *A. muricata* fruits, along with their respective $^1$H and $^{13}$C NMR data. The former ($^1$H NMR data) coincided faithfully with those observed in Figure 2. Although Hasrat et al. [6,7] does not describe the structure of isolaureline 4, it was possible to determine that this derivative contains an $OCH_3$ group attached at C-10 since the chemical shifts of H-8 and H-9 are shifted to higher field due to the donor effect of the $OCH_3$ group. In addition, multiplicity of H-9 (dd, *J* = 8.5, 1.9 Hz) corresponded to an *ortho* coupling with H-8 and *meta* with H-11. The rest of the functional groups were very similar to those present in annonaine 3. It was also found that isolaureline 4 was already described by Tomita and Kitamura [33] as a product of synthesis. Years later, this compound was isolated for the first time from a plant by Ziyaev et al. [34]. Although $^1$H and $^{13}$C NMR data of isolaureline 4 are not fully described to date, the NMR data collected in this paper are sufficient to propose that this alkaloid is also present in this mixture from *A. muricata* peel, since no such alkaloids have been reported in this raw material. For now, the present results motivate to apply the present protocol to extract, purify, quantify, and determine the structure and relative ratio of the minor and major alkaloids present in other organs of *Annona* species.

## 4. Conclusions

This study demonstrates the high efficiency of the ultrasound for extraction of alkaloids. However, the type and complexity of the plant material are important to obtain the optimal extraction conditions (amplitude and pulse cycle of ultrasound, and the resulting 5 min) of UAE. In addition, NMR-based spectroscopic experiments strongly support the presence of four main alkaloids, which were structurally identified as nornuciferine 1, assimilobine 2, anonaine 3, and isolaurelin 4 isolated from *Annona muricata* fruit peel. This is the first study where the presence of these alkaloids in the peel of *A. muricata* is reported. As confirmed by qualitative tests and spectroscopic analysis, ultrasound technology was shown to be an efficient innovative extraction technique to extract naturally occurring metabolites with nutraceutical and/or therapeutic interest.

**Author Contributions:** Conceptualization, G.A.-H., L.G.Z.-V., and E.M.-G. Methodology, G.A.-H., L.G.Z.-V., M.d.L.G.-M., M.I.G.-P., A.V.C.-T., and E.M.-G. Validation, G.A.-H., L.G.Z.-V., and E.M.-G. Formal analysis, G.A.-H., L.G.Z.-V., M.d.L.G.-M., M.I.G.-P., A.V.C.-T., C.R.-A., and E.M.-G. Writing—original draft preparation, G.A.-H., L.G.Z.-V., and E.M.-G. Writing—review and editing, M.d.l.Á.V.-V., A.P.-L., and C.R.-A. Project administration, L.G.Z.-V. and E.M.-G. Funding acquisition, L.G.Z.-V., A.P.-L., and E.M.-G. All authors have read and agreed to the published version of the manuscript.

**Funding:** The Tecnológico Nacional de México (Grant Number. 5089.19-P) and Secretaría de Investigación y Posgrado (SIP) del IPN (Grant Numbers 20180092 and 20194908) supported this work.

**Acknowledgments:** The authors gratefully acknowledge scholarship from CONACYT-Mexico to Gabriela Aguilar-Hernández. This work is part of the activities of the RED TEMATICA CONACYT 12.3 to Reduce and Valorize Food Losses and Waste: Toward Sustainable Food Systems.

**Conflicts of Interest:** The authors declare no conflicts of interest. The funders had no role in the design of the study, in the collection, analyses, or interpretation of data, in the writing of the manuscript, or in the decision to publish the results.

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
