# Peer review of "Extraction of Alkaloids Using Ultrasound from Pulp and By-Products of Soursop Fruit (Annona muricata L.)"

_applsci, doi:10.3390/app10144869_

Round 1

Reviewer 1 Report

The manuscript entitled “Extraction of alkaloids using ultrasound from pulp and by-products of soursop fruit (Annona muricata L)” by Aguilar-Hernández et al., compares ultrasound-assisted methods to improve the alkaloid extraction from different parts of the Annona muricata L. fruit.

The manuscript is in line with previous articles of the same group, describing the phenolic compound extraction by similar methods.

The text is well written; however, the manuscript could be improved. Please find attached a pdf with some highlighted parts that should be corrected.

The manuscript could be also improved by a brief discussion of the potential interest of the isolated alkaloids. For which applications could be used these alkaloids.

Author Response

Thank you very much for your accurate comments. We have done our best to follow up all your recommendations. Note: Changes in the revised manuscript are in red letters.

 Reviewer 1

  1. The manuscript is in line with previous articles of the same group, describing the phenolic compound extraction by similar methods.

The reviewer is right, we did a big project where the objectives were to use UAE to extract different metabolite secondaries on different parts of soursop fruit, the data were a lot; therefore, we consider to publish the data in different manuscripts. Although the methodology was the same, the results are very different; therefore, we consider that all the data could be published.

  1. The text is well written; however, the manuscript could be improved. Please find attached a pdf with some highlighted parts that should be corrected.

We corrected the errors highlighted in the pdf document.

  1. The manuscript could be also improved by a brief discussion of the potential interest of the isolated alkaloids. For which applications could be used these alkaloids.

We added in the page 14 lines 469-478, the discussion of the potential interest of the isolated alkaloids.

  1. English language and style are fine/minor spell check required

Our manuscript was checked by an expert English colleague and he did the pertinent corrections

Reviewer 2 Report

The work presents as main goal to measure the total alkaloid content (TAC) from pulp, peel, seed, and columella of soursop fruit (Annona muricata) by ultrasound-assisted extraction (UAE), and to obtain the best extraction conditions with the response surface methodology. Also, four main alkaloids were structurally identified. It is interesting to know the optimal extraction conditions for ALKs in the different parts of soursop fruit, taking into account that these compounds are generally present in plants in low proportion and the high complexity of the matrix. It was found that the TAC was higher in the peel (7.48 mg/g), and there is no reports on the content of ALKs from this part that is considered as a by-product. 

But, the authors have recently published two articles in MOLECULES for the extraction of phenolic compounds and for acetogenins from Annona muricata (Optimization of Ultrasound-Assisted Extraction of Phenolic Compounds from Annona muricata By-Products and Pulp. Molecules 2019, 24, 904; doi:10.3390/molecules24050904; and                Ultrasound-Assisted Extraction of Total Acetogenins from the Soursop Fruit by Response Surface Methodology. Molecules 2020, 25, 1139; doi:10.3390/molecules25051139, respectively). The present manuscript, as the previous two published works, are focused on the application of UAE and to study by RSM the optimal extraction conditions. It is surprising that only one of these previous articles is referenced in the manuscript, and only in the introduction to indicate that the soursop Annona muricata L. is a good source of bioactive compounds (Line 41). Comparing the three works, the methodology, approach and development, as well as part of the discussion, in this manuscript are very similar to the works published in MOLECULES.

On the other hand, although the structural characterization of the main alkaloids in peel from soursop is interesting, compounds 1,2 and 3 had been previously isolated and characterized by 1H-NMR and 13C-NMR as the main bioactive isoquinoline alkaloids from the fruit of Annona muricata (Hasrat et al, 1997). In the present work, an adequate isolation of the compounds would have been desirable for their characterization. On the other hand, it was indicated that the complete assignment of the signals for each alkaloid was performed by two-dimension NMR experiment by COZY and HSQC. But the results are not explained, nor is any additional information provided.

The reference of Hasrat et al. [5] included in the article is not correct. Thus, reference 5 “Hasrat, J.A.; De Bruyne, T.; De Backer, J.P.; Vauquelin, G.; Vlientinck, A.J. Isoquinoline derivatives isolated from fruit of Annona muricata as 5-HTergic 5-HT1A receptor agonists in rats: unexploited anti-depressive (Lead) products. J. Pharm. Pharmacol. 1997, 49, 1145-1149. 396 https://doi.org/10.1111/j.2042-7158.1997.tb06058.x” must be substitute by the appropriate reference “ Hasrat JA, Pieters L, De Backer JP, Vauquelin G, Vlietinck AJ. Screening of medicinal plants from Suriname for 5-HT (1A) ligands: Bioactive isoquinoline alkaloids from the fruit of Annona muricata. Phytomedicine. 1997; 4(2):133-140. doi:10.1016/S0944-7113(97)80059-1”.

Reference 28 is noted to describe isolaurelin 4, but this is difficult to determine because this reference is written in Chinese.

Author Response

Dear editor and reviewers:

Thank you very much for your accurate comments. We have done our best to follow up all your recommendations. Note: Changes in the revised manuscript are in red letters.

Reviewer 2

  1. The work presents as main goal to measure the total alkaloid content (TAC) from pulp, peel, seed, and 1. the authors have recently published two articles in MOLECULES for the extraction of phenolic compounds and for acetogenins from Annona muricata(Optimization of Ultrasound-Assisted Extraction of Phenolic Compounds fromAnnona muricata By-Products and Pulp. Molecules 2019, 24, 904; doi:10.3390/molecules24050904; and Ultrasound-Assisted Extraction of Total Acetogenins from the Soursop Fruit by Response Surface Methodology. Molecules 2020, 25, 1139; doi:10.3390/molecules25051139, respectively). The present manuscript, as the previous two published works, are focused on the application of UAE and to study by RSM the optimal extraction conditions. It is surprising that only one of these previous articles is referenced in the manuscript, and only in the introduction to indicate that the soursop Annona muricata is a good source of bioactive compounds (Line 41).

We added the reference Aguilar-Hernández et al (2020), and in the page 2 lines 79-83 we added the conclusions of reference. “Recently, Aguilar-Hernández [6, 7] found that UAE was effective in extracting polyphenols and acetogenins from soursop pulp and by-products, and concluded that, depending on the type of bioactive compound extracted and on each part of the fruit, UAE extraction conditions change because factors such as the physicochemical stability of the compounds and the extraction matrix are involved”

  1. Comparing the three works, the methodology, approach and development, as well as part of the discussion, in this manuscript are very similar to the works published in MOLECULES.

The reviewer is right, we did a big project where the objectives were to use UAE to extract different metabolite secondaries on different parts of soursop fruit, the data were a lot; therefore, we consider to publish the data in different manuscripts. Although the methodology was the same, the results are very different; therefore, we consider that all the data could be published.

  1. On the other hand, although the structural characterization of the main alkaloids in peel from soursop is interesting, compounds 1,2 and 3 had been previously isolated and characterized by 1H-NMR and 13C-NMR as the main bioactive isoquinoline alkaloids from the fruit of Annona muricata(Hasrat et al, 1997). In the present work, adequate isolation of the compounds would have been desirable for their characterization.

We appreciate the recommendation of the reviewer; however, we can say at least with our results it was demonstrated that the A. muricata peel can be a source of four alkaloids, because this raw material had not been studied.

Different solvents were used to purify the alkaloids by column chromatography, such as dichloromethane, ethyl acetate and methanol as a binary and tertiary mixture thereof in various proportions. However, the ratio (1:2:0.2 v/v) of the above-mentioned solvents proved to be the best combination for obtaining enriched alkaloid fractions. After this experience, we consider that HPLC will be the technique of choice to obtain alkaloids with high chemical purity for individual assignment of their respective 1H and 13C NMR spectra.

  1. On the other hand, it was indicated that the complete assignment of the signals for each alkaloid was performed by two-dimension NMR experiment by COZY and HSQC. But the results are not explained, nor is any additional information provided.

COSY and HSQC spectra were included in Figures 2 and 4, respectively (page 11, lines 336-341)). A brief discussion of representative signals of the identified alkaloids was also included.

  1. The reference of Hasrat et al. [5] included in the article is not correct. Thus, reference 5 “Hasrat, J.A.; De Bruyne, T.; De Backer, J.P.; Vauquelin, G.; Vlientinck, A.J. Isoquinoline derivatives isolated from fruit of Annona muricata as 5-HTergic 5-HT1A receptor agonists in rats: unexploited anti-depressive (Lead) products. J. Pharm. Pharmacol. 1997, 49, 1145-1149. 396 https://doi.org/10.1111/j.2042-7158.1997.tb06058.x” must be substitute by the appropriate reference “ Hasrat JA, Pieters L, De Backer JP, Vauquelin G, Vlietinck AJ. Screening of medicinal plants from Suriname for 5-HT (1A) ligands: Bioactive isoquinoline alkaloids from the fruit ofAnnona muricata. Phytomedicine. 1997; 4(2):133-140. doi:10.1016/S0944-7113(97)80059-1”.

The reference is correct, actually to exist two references of Hasrat et al., 1997a and b, we analyzed both and, we put two references in the reference list.

Reviewer 3 Report

The manuscript "Extraction of alkaloids using ultrasound from pulp and by-products of soursop fruit (Annona Muricata L)" explained to measure TAC by UAE method. Though the authors explained about the detailed techniques in this paper, need to give some detailed explanations.

  1. What about the effectiveness (%) of the UAE method and comparison with maceration extraction (ME)?
  2. Author mentioned about 2D NMR and 13C NMR spectra, it would be more supportive if the spectra would be included in the manuscript.
  3. Author characterized the 1D 1H NMR spectra, highlighted the residual signal of the CH2O of ethanol, what about the CH3 signal of ethanol?
  4. Did author try to get rid of the ethanol solvent peak from the sample?
  5. Author showed 3.8 - 2.6 ppm region in the 1H NMR spectrum correspond to H (4a,b and 5a,b). But these protons give doublet of doublet signal in the spectrum. There are no such signals in the spectrum.
  6. Author can try different eluting solvents to isolate the products in the column chromatography.
  7. Also, author can try recrystallization method to purify the compounds.

Author Response

Thank you very much for your accurate comments. We have done our best to follow up on all your recommendations. Note: Changes in the revised manuscript are in red letters.

Reviewer 3

  1. ¿What about the effectiveness (%) of the UAE method and comparison with maceration extraction (ME)?

We added the results in the manuscript, due to we did that experiment.

  1. Author mentioned about 2D NMR and 13C NMR spectra, it would be more supportive if the spectra would be included in the manuscript.

COSY, HSQC and 13C NMR spectra were included in the manuscript and they are briefly discussed.

  1. Author characterized the 1D 1H NMR spectra, highlighted the residual signal of the CH2O of ethanol, what about the CH3signal of ethanol.

The methyl group of ethanol appears as a triplet signal at d 1.24, and it is outside the spectral region (higher field) shown in Figure 2.

  1. ¿Did author try to get rid of the ethanol solvent peak from the sample?

Not in this case, because the residual ethanol signals (RMN 1H, d 1.24, t, J = 7.03 H, CH3, and d 3.72, q, J = 7.03 H, CH2) did not interfere with the assignment of the signals belonging to the identified alkaloids. As mentioned in the manuscript, further efforts are underway in our research group to achieve isolation, purification and structural characterization of the major alkaloids present in A. muricata. Part of this work will be to carry out the unambiguous assignment of the 1H and 13C NMR  spectra, as was recently described for other aporphine alkaloids (Ontiveros-Rodriguez et al, reference 21).

  1. Author showed 3.8 - 2.6 ppm region in the 1H NMR spectrum correspond to H (4a,b and 5a,b). But these protons give doublet of doublet signal in the spectrum. There are no such signals in the spectrum.

In fact, the multiplicity generated by scalar coupling between H4a,b and H5a,b is indeed more complex. If we consider only geminal and vicinal couplings, each proton of the CH2(4)-CH2(5) fragment would give as add signal. However, as you mention, this multiplicity is not perceived due to the strong overlapping of the signals generated for the mixture of the 4 identified alkaloids. Further, it can be expected a long-range scalar coupling of H4a,b with H6a, as was recently described in Figure 5 of reference 21.

  1. Author can try different eluting solvents to isolate the products in the column chromatography.

Thanks for the suggestion. Different solvents were used to purify the alkaloids by column chromatography, such as dichloromethane, ethyl acetate and methanol as a binary and tertiary mixture thereof in various proportions. However, the ratio (1:2:0.2 v/v) of the above-mentioned solvents proved to be the best combination for obtaining enriched alkaloid fractions. After this experience, we consider that HPLC will be the technique of choice to obtain alkaloids with high chemical purity for individual assignment of their respective 1H and 13C NMR spectra.

  1. Also, author can try recrystallization method to purify the compounds.

Thanks for the suggestion. It is an excellent alternative that often gives satisfactory results. We will try it out as part of achieving the next objectives of this research.

Round 2

Reviewer 2 Report

The changes carried out in the original manuscript have improved it. Nevertheless, the paragraph included from line 470 to 479 is not adequate for results and discussion and it could be moved to the introduction section.

Author Response

Dear editor and reviewers:

Thank you very much for your accurate comments. We have done our best to follow up all your recommendations. Note: Changes in the revised manuscript are in red letters.

Reviewer 2

The changes carried out in the original manuscript have improved it. Nevertheless, the paragraph included from line 470 to 479 is not adequate for results and discussion and it could be moved to the introduction section.

The paragraph from line 470 to 479, was moved to the introduction, Pag 2 lines 57-63, as reviewer recommended.
